



# Remote Sensing of *Trichodesmium* spp. mats in the Western Tropical South Pacific

**Guillaume Rousset[1], Florian De Boissieu[2], Christophe E. Menkes[3], Jérôme Lefèvre[4], Robert Frouin[5], Martine Rodier[6], Vincent Ridoux[7], Sophie Laran[7], Sophie Bonnet[8], and Cécile Dupouy[8]**

[1]IRD (Institut de Recherche pour le Développement), UMR ESPACE DEV, Nouméa, New-Caledonia

[2]IRSTEA, UMR TETIS, Montpellier, France

[3]IRD-Sorbonne Universités (UPMC, Université Paris 06)-CNRS-MNHN, Laboratoire d'Océanographie et du Climat: Expérimentations et Approches Numériques (LOCEAN), IRD Nouméa BP A5, 98848 Nouméa Cedex, New Caledonia

[4]LEGOS/MIO, 98800 Nouméa, New Caledonia

[5]Scripps Institution of Oceanography, University of California San Diego, La Jolla, CA 92093-0224, USA

[6]EIO (Ecosystèmes Insulaires Océaniens), Institut de Recherche pour le Développement-Université de la Polynésie Française-Institut Malarmé-Ifremer, Papeete, French Polynesia

[7]PELAGIS, UMS 3462, Université de la Rochelle/CNRS, La Rochelle, France

[8]Aix Marseille Université-CNRS-INSU, IRD, Mediterranean Institute of Oceanography (MIO), UM 110, IRD Nouméa, BP
A5, 98848 Noumea Cedex, New Caledonia

*Correspondence to*: Guillaume Rousset (guillaume.rousset@ird.fr)

**Abstract.** *Trichodesmium* is the main nitrogen-fixing species in the South Pacific region, a hotspot for diazotrophy. Due to the paucity of in situ observations, methods for detecting *Trichodesmium* presence on a large scale have been investigated to assess the regional-to-global impact of these species on primary production and carbon cycling. A number of satellite-derived
algorithms have been developed to identify *Trichodesmium* surface blooms, but determining with confidence their accuracy has been difficult, chiefly because of the scarcity of sea-truth information at time of satellite overpass. Here, we use a series of new cruises as well as airborne observational surveys in the South Pacific to quantify statistically the ability of these algorithms to discern correctly *Trichodesmium* surface blooms in the satellite imagery. The evaluation, performed on MODIS data at 250m and 1km resolution acquired over the South West Pacific, shows limitations due to spatial resolution, clouds, and
atmospheric correction. A new satellite-based algorithm is designed to alleviate some of these limitations, by exploiting optimally spectral features in the atmospherically corrected reflectance at 531, 645, 678, 748, and 869 nm. This algorithm outperforms former ones near clouds, limiting false positive detection, and allowing regional scale automation. Compared with observations, 80% of the detected mats are within a 2 km range, demonstrating the good statistical skill of the new algorithm. Application to MODIS imagery acquired during the February-March 2015 OUTPACE campaign reveals the presence of
surface blooms Northwest and East of New Caledonia and near 20°S-172°W in qualitative agreement with measured nitrogen fixation rates. The new algorithm, however, fails to detect sub-surface booms evidenced in trichome counts. Improving *Trichodesmium* detection requires measuring ocean color at higher spectral and spatial (< 250 m) resolution than MODIS,



taking into account environment properties (wind, sea surface temperature, …), fluorescence, and spatial structure of filaments, and a better understanding of *Trichodesmium* dynamics, including aggregation processes to generate surface mats. Such sub-mesoscales aggregation processes for *Trichodesmium* are yet to be understood.

# 1 Introduction

The Western Tropical South Pacific (WTSP) is a Low Nutrient Low Chlorophyll (LNLC) region, harboring surface nitrate concentrations close to detection limits of standard analytical methods, and limiting for the growth of the majority of phytoplankton species (Le Borgne et al., 2011). This lack of inorganic nitrogen favors the growth of dinitrogen ($N_2$)-fixing organisms (or diazotrophs), which have the ability to use the inexhaustible pool of $N_2$ dissolved in seawater and convert it into bioavailable ammonia. Several studies have reported high $N_2$ fixation rates in the WTSP (Berthelot et al., 2017; Bonnet et al.,

2009, 2015; Garcia et al., 2007), that has recently been identified as a hot spot of $N_2$ fixation (Bonnet et al., 2017). During austral summer conditions, $N_2$ fixation supports nearly all new primary production and organic matter export (Caffin et al., This issue; Knapp et al., This issue) as nitrate diffusion across the thermocline and atmospheric sources of N are < 10 % of new N inputs. The cyanobacterium *Trichodesmium* is one of the most abundant diazotrophs in our oceans (Capone, 1997; Luo et al., 2012) and in the WTSP in particular (Tenorio et al., accepted; Stenegren et al., 2017), where it has recently been

identified, based on cell-specific $N_2$ fixation measurement, as the major $N_2$-fixing organism, accounting for > 60 % of total $N_2$ fixation (Bonnet et al., This issue). One of the characteristics of *Trichodesmium* is the presence of gas vesicles, which provide buoyancy (van Baalen and Brown, 1969; Villareal and Carpenter, 2003) and help maintain this cyanobacterium in the upper ocean surface. *Trichodesmium* cells are aggregated and form long chains called trichomes. Trichomes then can gather into colonies called "puffs" or "tuffs," and these colonies can aggregate at the surface of the water and form large mats that can

extend for miles and were detected since James Cook and Charles Darwin's expeditions. During the southern austral summer, *Trichodesmium* blooms have long been detected by satellite in the region, mostly around New Caledonia and Vanuatu (Dupouy et al., 2000, 2011), and later confirmed by microscopic enumerations (Shiozaki et al., 2014).

Identifying the occurrence and the spatial distribution of *Trichodesmium* blooms and mats is of primary importance to assess their contribution to primary production and biogeochemical cycles regionally. However because of their paucity, scientific

cruises alone are not sufficient to achieve such goal and remote sensing completed by sea observations of mats, appears as the best alternate solution for assessing its global impact. By using specific optical properties of *Trichodesmium*, among which pigment absorption (mainly phycoerythrin, PE) and particle backscattering (Subramaniam et al., 1999a, 1999b), several bio-optical algorithms have been developed to detect *Trichodesmium* blooms in real time from various satellite sensors , i.e. the ones of Hood et al. (2002); Westberry et al. (2005); Dupouy et al. (2011) for SeaWiFS, the ones of Gower et al. (2014) for

MERIS, and the ones of Hu et al. (2010) and McKinna et al. (2011) for MODIS-Aqua (review in Mckinna (2015)).

The application of these algorithms to MODIS imagery revealed several issues, some of which had already been raised and discussed in the aforementioned articles. Atmospheric correction of satellite imagery above *Trichodesmium* mats is one of



these issues as reflectance from the floating algae can be wrongly interpreted as aerosols or clouds. It is a main concern in this region as the blooming period of *Trichodesmium* (mainly November to March (Dupouy et al., 2011)) coincides with the South Pacific Convergence Zone, i.e. heavy cloudiness making difficult the identification of coincident in-situ mats in satellite imagery. The local aggregation and small width of *Trichodesmium* mats (~ 50 m typically) also calls into question the influence

of ocean color sensors resolution on the detection quality of these mats.

The aim of this study is to manage a systematic detection of *Trichodesmium* blooms in the vast WTSP Ocean between latitudes 26° and 10°S and longitudes between 155° and 190° E, building on previously published algorithm and, in particular, provide an updated *in situ* database on which these algorithms can be evaluated with newly acquired datasets and particularly during the "Oligotrophy from Ultra-oligoTrophy PACific Experiment"(OUTPACE) cruise (DOI:

http://dx.doi.org/10.17600/15000900) in March-April 2015 (Moutin et al., 2017). To achieve this objective, a large database of mat observations in this region was created in order to evaluate retrievals from MODIS reflectance. Because of the specificities of the MODIS latest release and the presence of numerous clouds in the WTSP, the existing algorithms had to be adapted whenever possible. From this experience, a new algorithm was then created for the detection of *Trichodesmium* in the WTSP, more robust to cloud cover and tested on high resolution MODIS imagery, building on the algorithms of (McKinna et

al., 2011) and (Hu et al., 2010). The paper is organized as follows: Section 2 presents in-situ data and satellite image used in this study. Section 3 covers the methods used in order to extract *Trichodesmium* spectral signature and their limitations, as well as the three detection algorithms evaluated in this study. Section 4 presents the detection performances of two former compared to the newly developed algorithm and the detection results this last algorithm on the OUTPACE cruise path. Section 5 discusses the new algorithm performances. Section 6 draws the conclusions and perspectives of this study.

**2 Material**

**2.1 In situ observations**

The in-situ database used to train and test the *Trichodesmium* detection model is a combination of three datasets intersecting with the MODIS acquisition period (2000-present). It includes the *Trichodesmium* mat observations published in Dupouy et al. (2011). These observations were done between 1998 and 2010, from aircraft, French Navy ships, research vessels (e.g. R/V

Alis) and ships of opportunity. Some of these visual observations were confirmed by water samples analyzed with photomicrographs confirming the presence of abundant *Trichodesmium* (Dupouy et al., 2011). Airborne visual observations were also gathered in December 2014 in the vicinity of New Caledonia during the REMMOA program (Laran et al., 2016). This second dataset provides a large number of *Trichodesmium* mat observations along numerous and repetitive transects, which is most favorable for satellite data validation. During the same period, several in-situ observations of mats were made

during the SPOT 4 scientific cruise (Biegala et al., 2014), coincident with MODIS imagery and thus constituting a third dataset. In total, the database created from the compilation of these open ocean observations contains 507 observations (Figure 1) in the region 15°S-25°S, 155°E-180°E. It is referred to as the Simple Observation Base (SOB) in the following.


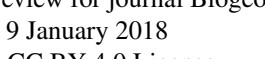


In addition to SOB, a latitudinal transect around 20°S was carried out during the OUTPACE scientific cruise (Moutin et al., 2017) covering the region 160°E-160°W from February 23th to April 1st in 2015. Seawater samples were collected for *Trichodesmium* quantification by quantitative PCR as described in Stenegren et al. (2017), microscopic counts at selected stations (Caffin, Comm. Pers.), as well as N$_2$ fixation rates as described in Bonnet et al. (This issue). Moreover, *Trichodesmium*

abundance from the Underwater Vision Profiler 5 (UVP5; (Picheral et al., 2010)), calibrated on trichome concentration from pigment algorithms and on visual counts of surface samples at all stations allowed to describe the distribution of *Trichodesmium* along the transect (Dupouy et al., This issue).

## 2.2 Satellite imagery

The satellite data used in this study was all MODIS-Aqua and MODIS-Terra data corresponding in time and space to the SOB

database and the OUTPACE campaign. Image were downloaded at level L1A (oceandata.sci.gsfc.nasa.gov) and processed with SeaDAS v7.0.2 to produce a L2 MODIS Data basis at 250 m and 1 km resolution. Standard SeaDAS masks were applied during this processing: atmospheric correction failure, land, sunglint, very high or saturated radiance, sensor view zenith angle exceeding threshold, stray light contamination and cloud contamination. In order to reduce the influence of these phenomena, only the observations with less than 60 % of mask coverage over a 0.5° radius area around point location were kept.

We used MODIS radiance in visible, near-infrared (NIR) and short wavelength infrared (SWIR) at different resolutions: 250 m resolution for bands 1 (645 nm) and 2 (859 nm), 500 m resolution (bands 3-7, visible and SWIR land/clouds dedicated bands) and 1 km resolution (bands 8-16, VNIR ocean color dedicated bands).To evaluate the influence of resolution on detection performances, L2 remote sensing data was produced at both 250 m and 1 km resolutions, with interpolation of 500 m and 1 km channels and aggregation of 250 m resolution channel respectively. The consequences of these processing are

discussed in Section 5. The images were processed with the SeaDAS standard atmospheric correction which is a correction using bands 15 (748 nm) and 16 (869 nm). The method NIRSWIR (Wang and Shi, 2005) for aerosols correction was used specifically for detection algorithm of (McKinna et al., 2011) as specified in the same publication.The aerosol correction has consequences on floating algae reflectance retrieval, as it is shown in the following section. In order to avoid this kind of issues, the reflectances were also computed without the aerosol correction, which avoids the wrong aerosol correction and the

eventuality of negative reflectance values as suggested by Hu et al. (2010). The remote sensing reflectances after Rayleigh and standard aerosol correction (Rrs), and the reflectance corrected only by the Rayleigh effect (Rrc), are the MODIS L2 products used for this study. Downloading and processing of MODIS images was based on a ProsgreSQL satellite database and a python software developed for this study and ready for real-time *Trichodesmium* reporting during the OUTPACE cruise.





## 3 Methods

### 3.1 Atmospheric correction: sensitivity and adjustment

Atmospheric overcorrection is a general problem for strong floating algae concentrations and have been noticed repeatedly (Hu et al., 2010). A major hypothesis of atmospheric correction and cloud detection algorithms is to consider seawater as a

black body in the NIR and SWIR. However, *Trichodesmium* mats floating at the surface present a strong reflectance in these wavelengths due to chlorophyll pigment (red-edge). Thus it is wrongly interpreted as aerosol by the atmospheric correction algorithms. It results in reflectance values excessively reduced, even leading to negative values in some cases, and has consequences on all further computed L2 products.

This phenomenon is illustrated in Figure 2 presenting a MODIS-Aqua image of the Australian coast acquired just after a period

of heavy rain that led to a massive *Trichodesmium* bloom. Fortunately, this bloom could also be observed in-situ (McKinna et al., 2011). Figure 2A shows the "true-color" image obtained by combination of Rrc's. On this image, large visible *Trichodesmium* mats distributed over a vast area can be seen. Figure 2B displays the aerosol optical thickness (AOT) at 555 nm, an indicator of the aerosol load in the atmosphere. The high values of AOT match the filament spatial structure noticed in the "true color" image. However, this spatial organization is quite unlikely to be due to aerosol structures, as they are very thin

and do not seem to be driven by wind. Moreover, the center of the blooming regions is masked (grey patches on figure 2B), although the "true color" image does not indicate the presence of clouds in this particular area. These pixels were wrongly identified as cloud by the cloud detection algorithm because of SWIR Rrs values higher than 3% (Wang and Shi, 2005). Figure 2C shows the chlorophyll concentration estimated according to the OC3 algorithm (Hu et al., 2012). Chlorophyll concentration decreases systematically, even falling to zero, in the vicinity of the *Trichodesmium* distribution patterns, although the real

concentration is certainly large and higher at the core of the mats than at its periphery. The spectral signature of the mats are studied in more details in next section, to show the consequences of the miscorrection on further used detection algorithms and the interest of replacing Rrs by Rrc for floating algae detection.

### 3.2 Extraction of the spectral signature of mats

With the regular cloud cover of the region, the number of strict coincidences of in-situ observations and cloud free MODIS

images where *Trichodesmium* mats are visible is small. Therefore, the search for coincidences has been extended in space and time. To extract *Trichodesmium* spectral signature, 6 tiles have been specifically selected (Table 1) and are used in order to test the different bio-optical algorithms designed to detect the *Trichodesmium* presence. These images have been chosen because they are mostly clear (i.e., contain few clouds), *Trichodesmium* mats are visible in the "true color" images, and numerous in situ observations exist in the entire area (Figure 3).

The NASA method which allows one to select match-ups, i.e., average or nearest pixel, has been used to find coincidences between in situ observations and clear MODIS satellite pixels (Bailey and Werdell, 2006). A total of 468 satellite pixels were found coincident to the SOB database. Only 50 remain after the mask application. Thus, approximately 90 % of in-situ



observations are not usable, mainly because of cloud cover. Once inspected manually and sorted out, 19 spectra out of the 50 pixels selected exhibit fluctuations similar to the *Trichodesmium* signal presented in Hu et al. (2010) and McKinna et al. (2011). In order to increase the number of useful observations, the coincidence detection was extended to a temporal window of +/- 4 days and the search area up to +/- 50 km (200 pixels at 250 m resolution) considering that the drifting speed of the algae mat

could be up to 0.5 m/s when the weather condition is favorable, i.e. wind speed sufficiently low to keep *Trichodesmium* aggregates at surface. Also, some in-situ observations close spatially and temporally (in the same tile and at intervals of +/- 4 days) increased our degree of confidence in identifying the filamentous patterns as *Trichodesmium*.

From the 6 MODIS tiles, two types of spectral signatures have been identified and extracted. The first type is the signature of high algae concentration. With the hypothesis that only the algae *Trichodesmium* can make floating algae bloom in WTSP

region, pixels have been selected when there was a high Floating Algae Index (FAI) (Hu, 2009), visible mats on "true color" image and remote sensing chlorophyll concentration anomaly. The second type of spectral signature is from areas selected immediately next to the *Trichodesmium* mats, i.e., a pixel next to a FAI-selected pixel and without visible algae. These were selected for each resolution of the satellite sensor. Indeed, a high probability of high concentration of mixed or deeper *Trichodesmium* colonies in the water column is expected in these areas. In the end 1200 spectra were extracted, with 600 examples for each case.

Figure 4 presents the average and standard deviation of the Rrs and Rrc spectra of mats and adjacent to mats pixels. The standard atmospheric correction of SeaDAS (two-bands correction) has been applied for the Rrs spectra, which leads to systematic zero value at the wavelengths used to calibrate correction, i.e. band 15-16 (748 and 869 nm). One can notice that Rrs and Rrc average spectra have similar shapes, with an offset, keeping Rrc positive at all wavelengths. In comparison with

*Trichodesmium* spectrum of the literature, i.e. the ones in Hu et al. (2010) or in McKinna et al. (2011), Figure 4 shows that several similarities appear. Spectra are showing a strong negative slope in the visible channels (from 400 to 600 nm) and a "red-edge" more pronounced for mat pixels than for adjacent pixels. Negative values of Rrs are occurring at 678 nm (maximum of chlorophyll absorption) and at 859 nm. Comparison between Rrc and Rrs shows interestingly that standard deviation error bars are much smaller for Rrc reflectances while the range of magnitudes between wavelengths is larger. This is a significant

argument for using Rrc instead of Rrs, as it would lead to a better discrimination of *Trichodesmium* mat spectra against other spectra.

If the negative slope at 678 nm can still be seen at 1 km resolution (Figure 4E), the negative spectral gradient at 859 nm observed on pixels adjacent to mats on 250 m resolution data was not noticed. This issue has already been noticed by Hu et al. (2010), the negative slope seems to be generated by the interpolation process while upscaling from 1 km to 250 m.

**3.3 Two published algorithms**

In order to detect mats (Gower et al., 2014) used the remote sensing chlorophyll concentration the 700 nm channel, which is a key factor of his algorithm. Unfortunately, this band present on SeaWiFS is missing on MODIS. Thus, from all the

*Trichodesmium* bloom detection algorithms with MODIS, only the algorithms of McKinna et al. (2011) and Hu et al. (2010), designed for the MODIS sensor, have been implemented and tested.

The *Trichodesmium* detection algorithm of McKinna et al. (2011) is based on 4 criteria relative to the shape of Rrs (see definition in Appendix). When we applied it on the same MODIS image as the one used by McKinna et al. (2011), the detection results of this algorithm showed more disregarded pixels because of the 4th criterion (eliminating pixels which have a negative magnitude of nLw at wavelength 555, 645, 678 or 859 nm). Indeed, the test of a negative Rrs value at 678 nm due to aerosol overcorrection excludes many pixels. Skipping the 4th criterion of the algorithm allowed to match the results of McKinna et al. (2011). Therefore, this modification was adopted for this study and the algorithm is called "McKinna modified" in the following.

The *Trichodesmium* detection algorithm presented in Hu et al. (2010) is based on a two steps analysis of Rrc spectra, 1) identify of strong floating algae concentrations with FAI, 2) resolve ambiguity between algae species analyzing the spectral shape, i.e. *Trichodesmium* and *Sargassum*. To avoid spectral influence of eventual aerosols, Hu et al. (2010) propose a correction method simply based on the difference of Rrc spectra between bloom and nearby algae free region. After several try on the data presented above, this correction method was found to be sensitive to the choice of the algae free region (not shown in this article). Thus, we kept only the first step of his algorithm (FAI) and apply a threshold between 0 and 0.04 to detect the algal signal in the following.

**3.4 New algorithm criteria**

Our criteria for detecting *Trichodesmium* mats were defined based on spectral characteristics of Rrs and Rrc (Figure 4). Indeed, the systematic negative Rrs values at 678 nm over strong *Trichodesmium* mat concentrations is taken as an advantage here. All pixels with negative Rrs value at this wavelength have a high probability to be floating algae and thus *Trichodesmium* in this region. The absolute value of Rrs(678) is actually used as an index of mats concentration, and can also be used to detect some artifact, e.g., sun glint (Eq. 1).

Similarly to the algorithms of Hu et al. (2010) and McKinna et al. (2011) (appendix A), three criteria were defined to extract the typical spectrum shape of *Trichodesmium*: 1) Rrs(678), as the spectrum shape may be affected by the aerosol miscorrection of SeaDAS standard atmospheric correction algorithm in the presence of mats (Eq. 1); 2) Rrc(748) and and Rrc(8679) are used to detect the presence of the red-edge associated with the surface *Trichodesmium* mats, which is one of the main criteria (Eq. 2); and 3) Rrc(645) and Rrc(531) are used to resolve ambiguities between *Trichodesmium* mats and incorrectly detected pixels after processing with previous criteria, the misdetections occurring mostly in cloud neighborhood (Eq. 3).

$$Rrs(678) < 0 , \tag{1}$$

$$Rrc(748) < Rrc(859) , \tag{2}$$

$$Rrc(645) < Rrc(531) , \tag{3}$$



## 4. Results

### 4.1 Algorithm application

An attempt to compare efficiency of the three *Trichodesmium* detection algorithms is illustrated in Figure 5 on the MODIS tile A2007290.0355, used in McKinna et al. (2011). The McKinna modified algorithm shows the same detection patterns as the

ones found in McKinna et al. (2011). It is a vast area of *Trichodesmium* within which the filamentous structures cannot be distinguished. The new algorithm and the FAI (with a threshold between 0 and 0.04) show thin filamentous structures more similar to *Trichodesmium* mat structures observed on airborne photographs. The new algorithm provides values which are the amplitudes of the negative correction (at 678 nm).

Compared with both former algorithms, the new algorithm performs much better near clouds. Figure 6 is a zoom of the red

rectangle of Figure 5. This area presents a cloud path where McKinna modified algorithm and Hu modified algorithm detect *Trichodesmium* pixels. These pixels were identified as false positives as their spatial distribution is sparse and only in the vicinity of clouds. This conclusion is also supported by the "true color" composition (Figure 2) where the only *Trichodesmium* mats seem to be the ones at the bottom of the image. In that area the new algorithm does not make any false positive detection while keeping the *Trichodesmium* mats at the bottom of the image. The robustness of the new detection algorithm to clouds

while keeping accurate *Trichodesmium* mat detection is an important improvement for regions with high cloud covering, such as the WTSP.

### 4.2 Algorithm performance and comparison with in-situ mat observations

The exact coincidence in time and space between in-situ *Trichodesmium* mats observations and satellite mat detection is quite difficult to reach in general. One of the main reasons is by far the cloud cover, which eliminates a large quantity of the possible

comparisons (90 %). A second reason is the elapsed time between in-situ observations and the corresponding satellite pass during which the floating algae could have drifted at sea surface and/or migrated vertically depending on sea conditions (temperature, wind, etc.). For example, the abundance of *Trichodesmium* at the sea surface may vary with the time of day, as a daily cycle of rising and sinking of colonies in the water column is often observed as a result of cell ballasting (Villareal et al., 2003). Moreover, as *Trichodesmium* acts as buoyant particle, it can be advected by surface currents. Given the highest

surface current speeds, such as ~ 0.5 m/s at most in eddies, a mat would have drifted by ~ 50km in a day and is unlikely to escape the satellite acquisition area. However, that is a worst case scenario as eddies in that regions generally have speed lower that 0.3 m/s, (Rousselet et al., This issue)(Cravatte et al., 2015).

To circumvent that problem and present a more statistically robust comparison of our algorithm with in situ data, we used the

following strategy. With the hypothesis that a bloom can last for ~one week (e.g Kumar et al., 2015), an analysis of the spatio-temporal distance between the closet in-situ observation and the nearest detected mat was conducted. For each day in a range





of +/- 4 days around the date of observations, the spatial distance between the position of the observation and the nearest detected mat was computed.

Figure 7 presents the spatio-temporal results obtained with the new algorithm, by distance intervals of 0.5 km. It shows that the proportion of coincidences decreases with the distance, which was the expected behavior as changes in environmental

conditions are increasing with distance. It also shows that there is a high probability to find a mat near the location of an in-situ observation independently of the number of days that separates the observation from the tile acquisition. Overall, 80 % of the observed mats have a corresponding mat detection within less than 2 km range. These results demonstrate the statistical capability of the new algorithm to retrieve a mat near a point of observation.

## 4.3 Algorithm application for the OUTPACE cruise

The new algorithm was applied to MODIS data at the OUTPACE cruise time. A total of 140 tiles at 250 m resolution were covering the time period (2015-02-15 to 2015-04-07) and the spatial area of the cruise. Due to an important cloud cover during the cruise, only a few tiles were exploitable. *Trichodesmium* mats were detected from 12 MODIS-Aqua and 3 MODIS-Terra tiles. Figure 8B shows the detected mats over these tiles (in cyan), superimposed. It is interesting to note that the OUTPACE cruise actually crossed a number of *Trichodesmium* satellite detections. In order to further illustrate the results, a crude

qualitative presence/absence scheme is performed to better visualize which OUTPACE stations were coincident with the algorithm detection. We selected areas within 50 km off each OUTPACE stations and labeled the station as presence when there was at least one pixel detected as positive in the satellite algorithm. In figure 8B, red points are presence, blue points are absence.

*Trichodesmium* mats were mostly observed visually northeast of New Caledonia one week before the cruise and during the

first days of the cruise (on board observation) by video and photographs. There was no other mats observed during the remainder of the cruise but there was not any dedicated observer that would actually permit such visual observation, unlike during the REMMOA campaigns. Nevertheless, UVP5 counts of colonies, phycoerythrin and trichome concentrations along the transect show that *Trichodesmium* contribution was maximum in the Melanesian Archipelago, the Western part of the transect (Dupouy et al., This issue), where slicks are numerous, and then fairly well related to *Trichodesmium* concentrations in

the upper layer. The other high spot of mats is at LDB, where no slick was observed but where *Trichodesmium* was in high concentration, although mixed with a high abundance of picoplanktonic cyanobacteria (Dupouy et al., This issue).

Bonnet et al (This issue) reported a significant ($p<0.05$) correlation between $N_2$ fixation rates and *Trichodesmium* abundances during OUTPACE. Bulk and cell-specific 15N2-based isotopic measurements, that *Trichodesmium* accounted for >80 % of $N_2$ fixation rates in this region at the time of the cruise. Such a high correlation between *Trichodesmium* biomass (here

phycoerythrin) was also measured in New-Caledonia waters (Tenorio et al, accepted). Hence the in situ $N_2$ fixation rate measured during the cruise (Figure 8A) is used as a robust proxy of the *Trichodesmium* concentration to further evaluate accuracy of satellite detections. A qualitative comparison between Figures 8A and 8B allows to see that when significant fixation rates were observed, *Trichodesmium* presence was detected by satellite and when the fixation rates were low

*Trichodesmium* absence was stated by the aforementionned algorithm. Although qualitative, this successful validation gives confidence in using our algorithm for *Trichodesmium* detection.

## 5 Discussion

### 5.1 Algorithm limitations

Even with a very strong algal concentration, it is possible that with oceanic weather conditions such as sufficient wind, *Trichodesmium* scatters and mixes vertically, i.e., we lose the strong signal in the infrared due to the red-edge linked to mats. We are then in the presence of *Trichodesmium* concentrations that cannot be detected completely with our algorithm. It is successful to locate highly concentrated surface mats, but is not suited for revealing *Trichodesmium* when scattered under the
surface. These are successful to locate the surface mats, but do not succeed in revealing *Trichodesmium* filaments and/or colonies when they are not aggregated in sea surface mats. We would need, in such situations, a new algorithm, which would allow estimation of *Trichodesmium* abundance over the whole upper layer. By examining the Rrs spectra of scattered *Trichodesmium*, obtained during OUTPACE and other cruises, it was not possible to identify clearly characteristics allowing *Trichodesmium* detection. We find ourselves dealing with a complex problem and a number of variables that, with our current
knowledge, do not allow us to create a new bio-optical algorithm and identify robustly *Trichodesmium* below the surface. (Dupouy et al., This issue) found that normalized water-leaving radiances in the green and yellow during OUTPACE were not totally linked to chlorophyll concentration unlike during BIOSOPE, which was hypothesized might result from an extra factor related to colony backscattering or fluorescence.

Considering the spatial and spectral resolution of the sensor MODIS, our algorithm optimizes the balance between
*Trichodesmium* detection and false positive. The new algorithm first criterion is a threshold that could be adapted. Here with the sensor MODIS, the negative values of the Rrs at 678 nm has been used as a spectral form criteria similar to the one used in (McKinna et al., 2011) was not enough to distinguish *Trichodesmium* from the rest. However, this criterium is fundamentally a nonsense as reflectance cannot be negative. Moreover the zero threshold has been chosen qualitatively and implies that it would have to be adjusted again in order to work elsewhere.
The algorithm has been designed and tested in the WTSP, but the literature provides only in-situ *Trichodesmium* spectra in other regions. Hence the satellite spectra retrieved (Figure 4) cannot be compared with coincident in situ spectra which were not acquired in our region. More precisely in the visible domain, spectra by McKinna and Hu are different from the ones retrieved in the WTSP, where spectra show a high disturbance between 412 and 678 nm in the literature, the fluctuation are close to the water signal in Figure 4. As the algorithm has been built from these spectra, it may be that others spectral shapes
are more pertinent in others areas. Finally, as this study has been carried out in the WTSP area, the robustness of this algorithm in the presence of other floating algal (e.g *Sargassum*) is also unknown.





As seen previously from the spectral view MODIS lacks several interesting band (Gower et al., 2014) that could be used in identifying *Trichodesmium*. From the available bands, we constructed our detection criteria leading to our second and third criteria. However the physical understanding of the phenomena behind our criteria are still unknown. Understanding the significance of our choices from the inherent optical properties of *Trichodesmium* should be undertaken for these criteria.

One should notice that only the densest mats of *Trichodesmium* are detected with this algorithm. The goal was to provide an algorithm that could detect automatically *Trichodesmium* in a global scale, and thus limiting the false positive detection as best as possible. Finally, the new algorithm is unable to determine the existence of thin superficial slicks and diffuse Trichodesmium in the water column. *Trichodesmium* quantification carried out during the OUTPACE campaign (Stenegren et al., 2017) revealed high *Trichodesmium* abundances near the Fiji island, while our algorithm did not detect them (Figure 8).

**5.2 Spatial resolution impact**

As indicated previously, only few spectral bands (land channels) have a high resolution (250 m or 500 m), while the rest have a resolution at 1 km. To investigate the influence of resolution on the spectral signature of *Trichodesmium* mats the spectral analysis was also conducted at a 1 km resolution. Dense groups of extended mats are still well detected at 1 km resolution. However, thinner mats with a weaker signal visible at 250 m resolution are lost at 1 km resolution. Figure 9 illustrates this

behavior on MODIS data.

The spatial structure of *Trichodesmium* aggregates is complex. When mats are present, *Trichodesmium* have a tendency to form a filamentous pattern much narrower than 250 m (50 m at most, according to visual detections), and thus the satellite sensor at 250 m resolution can only detect the largest ones (Figures 9 and 10). There is hence a scale mismatch between the exact form of the thin filaments and the actual detection by the current satellite data, which must average in a way the thin and

strongest filaments into signals detectable at 250 m. Understanding the shape of the filaments, and their physical characteristics (e.g width) will require much higher resolution satellite date (at least 50 m) which are available at present but without repetitive coverage. Figure 10 additionally illustrates that the *Trichodesmium* filaments are but a tiny part of the chlorophyll tongues and are inserted into the much wider chlorophyll patterns. There can be, within a chlorophyll tongue such as Figure 10, several thin elongated filaments.

One would also intuitively believe that the filaments illustrate the presence of dynamical fronts where convergent dynamics can maintain and participate to the mat aggregations. A natural dynamical criteria allowing to characterize the presence of the filaments could be found in the FSLE methods (Rousselet et al., This issue) but we could not associate the presence of the FSLE with the presence of the filaments, for instance on Figure 10 (not shown). Rousselet et al. (This issue) discuss the fact that FSLE only matched in situ chlorophyll "fronts" during OUTPACE with a 25 % correlation but we have seen that our

filaments are present at a scale finer that the chlorophyll scale detected by the satellite during OUTPACE (see also Figure 10). Our filaments are typically present at the sub-mesoscales, and we believe that it is unlikely that the present calculation of FSLE, using 12.5 km satellite data at best (Rousselet et al., This Issue) can in fact be used to understand the filament dynamics. If FSLE are the right tools to understand filament formation, they must be calculated using a much higher spatial resolution





than presently available. Hence, we lack the tools at present with which to understand the organization of the detected filaments and dedicated in situ experiments will have to be specifically undertaken to resolve that question.

## 6 Conclusions and perspectives

At present, previously published algorithms detecting *Trichodesmium* data (Hu et al., 2010; McKinna et al., 2011) using the current MODIS data archive, cannot be directly used to detect *Trichodesmium* mats automatically in the South Pacific as they either miss the mats due to algorithms failures (Section 3.3) and/or do not eliminate numerous false positive in the presence of clouds. In our paper, we have devised a new algorithm building on the previous ones, which allows a cleaner detection of those mats. One of the strengths of our study is the validation of our method with a new, updated database of mats in the South Pacific. This algorithm can however detect only the densest slick but achieves the goal of limiting the detection of false positive due to clouds. During the OUTPACE cruise, we show that satellite detections could help to confirm the presence of *Trichodesmium* slicks at much wider spatial range than what is possible to observe from a ship. Which illustrate the important contribution of satellite observations to seawater measurements. Yet, the new detection algorithm was developed and evaluated on WTSP region. Hence, future prospects will be to extend the evaluation to other regions, especially in the presence of other floating algae such as *Sargassum*.

MODIS-Terra and MODIS-Aqua satellite sensors are acquiring data since 2000 and 2002 respectively. However, the data quality of these sensors is becoming more and more uncertain with time going by, as their mission was not expected to last more than 6 years. The new algorithm could be adapted to other satellite instruments with similar spectral bands, for example VIIRS onboard NPP and NOAA-20 (1 km resolution) and OLCI onboard Sentinel-3 (300 m spatial sampling), but the spatial resolution remains a problem as we observed that 250 m was already to coarse a resolution to understand the thinner mat dynamics. A study with a better spectral and spatial resolution may lead to better performances and to a new and better algorithm, and this may be possible, at least regarding spatial resolution, with MSI onboard the Sentinel-2 series (10 to 60 m resolution).

It has been previously seen that near dense *Trichodesmium* mats, some product like the satellite chlorophyll concentration are erroneous. However in order to better constrain the contribution of *Trichodesmium* to nitrogen and carbon biogeochemical cycles, this algorithm must be corrected. The use of the Rrc instead of the Rrs is possible but some adjustments and comparisons with in-situ measurements must be carried out before proposing such algorithm. Globally this algorithm allows one to estimate the *Trichodesmium* aggregated in sea surface mats. The next step is to understand the quantitative aspect linking the *Trichodesmium* abundances to $N_2$ fixation rates, including their vertical distribution even when *Trichodesmium* filaments/colonies are spread out in the water column. Another important field of interest is to be able to understand phytoplankton functional types using satellites including *Trichodesmium* (de Boissieu et al., 2014). At present, we do not know any such study that included *Trichodesmium* but we have hopes that with our new in situ database and our understanding of

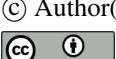



the mat shapes detected in the present study, and the development of performing statistical methods such as machine learning, advances can be made in that that regard. This will be undertaken in the future.

Finally (Dutheil et al., This issue) explore the regional and seasonal budget of the $N_2$ fixation due to *Trichodesmium* in a numerical model based on physical and biogeochemical properties that does not take into consideration the part of *Trichodesmium* that aggregates in mats. One interesting aspect will be to find a way to integrate our results in such model to better estimate the regional effects of that species.

## Appendix

### (McKinna et al., 2011) algorithm

The McKinna et al. (2011) algorithm is based on the analysis of the reflectance spectrum of a moderate Trichodesmium mat taken above the water, similar to the one measured on colonies in a small dish with an Ocean Optics spectroradiometer (Dupouy et al., 2008). It uses typical spectral characteristics of the normalized water-leaving radiance (nLw) after atmospheric correction to define 4 *Trichodesmium* detection criteria. The first three criteria relate to the shape of the spectrum and are given by the last criteria discards any pixel with negative nLw. When these 4 criteria are respected the pixel is identified as *Trichodesmium*:

$$nLw(859) > c_1 nLw(678) \tag{4}$$
$$nLw(645) > nLw(678) \tag{5}$$
$$nLw(555) > nLw(678) \tag{6}$$
$$nLw(555), nLw(645), nLw(678), nLw(859) < 0 \tag{7}$$

### (Hu et al., 2010) algorithm

Another detection algorithm, originally developed by (Hu, 2009) for floating algae, can be applied to *Trichodesmium* mats, as demonstrated by Hu et al. (2010) on MODIS-Aqua images of the west coast of Florida and the Gulf of Mexico, even though the *Trichodesmium* mats occurred in Case 2 waters. This algorithm can be decomposed into two steps: 1) detection of floating algae (FAI, Floating Algal Index), and 2) test of the form criteria of the radiance spectrum.

The FAI aims at detecting the strong reflectance in the infrared (red-edge) characteristics of the algal agglomerate at the ocean surface. To avoid the atmospheric overcorrection linked to the red-edge effect of the floating algae organized in a heap (Hu, 2009), the calculation of this index is applied to reflectance corrected only for the effects of Rayleigh scattering (Rrc). This correction accounts for the major part of the color of the atmosphere if aerosols are not too abundant (i.e., small optical thickness). The FAI is then defined as the difference between Rrc of the infrared domain (859 nm for MODIS) and a reference reflectance (Rrc0) calculated by linear interpolation between the red and shortwave infrared domains, respectively 667 nm and 1240 nm for MODIS:





$$FAI = R_{rc,NIR} + (R_{rc,SWIR} - R_{rc,RED}) \times \frac{(\lambda_{NIR} - \lambda_{RED})}{(\lambda_{SWIR} - \lambda_{RED})} \qquad (8)$$

$$\lambda_{RED} = 645\ nm\, , \lambda_{NIR} = 859\ nm\, , \lambda_{SWIR} = 1240\ nm \qquad (9)$$

where $_{RED}$ = 645 nm, $_{NIR}$ = 859 nm, and $_{SWIR}$ = 1240 nm. According to Hu et al. (2010), the difference between $R_{rc}$ and $R_{rc0}$

5 (the second term of Equation 8) allows one to deal with the majority of the atmospheric effect which has a quasi-linear spectral shape between 667nm and 1240nm.

The second step of the algorithm consists in identifying the mats emphasized by the FAI thanks to the shape of the spectrum in the visible domain. So as to correct the bias inferred in the visible part of the spectrum by the possible presence of mats, Hu et al. (2010) suggests applying to the pixels presenting a strong value of FAI, the correction of an area situated immediately

10 next to this pixel and without bloom. This approach being very expensive in times of calculation, it is substituted by a simple difference between the spectrum Rrc of the pixels suspected and that of a nearby zone without mat. The spectrum of difference of Rrc of *Trichodesmium* presents a pattern (spectral signature) that seems to be specific to it, i.e., a succession of high type low - top - low - top for the wavelengths 469-488-531-551-555 nm.





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

FIGURE LEGENDS

**Figure 1: Map of in-situ (visual) observations of *Trichodesmium* mats gathered on the studied region.**

**Figure 2: MODIS-Aqua tile A2007290.0355 used in McKinna et al. (2011): (A) Rrc "true color" composition; (B) aerosol optical thickness at 555nm; (C) Chlorophyll concentration product computed from Rrs.**

**Figure 3: "True color" image of the 17th December "2014A2014351.0255" for which in situ observations (red crosses) exist in the SOB database, and used for the test and adjustment of the different bio-optical algorithms.**

**Figure 4: MODIS Spectra of *Trichodesmium* mats (A and B) and adjacent areas (C, D and E) normalized by the maximum spectral value of all wavelengths, with pixels resolution at 250m (A, B, C and D) and 1km (E). B and D are Rrc reflectances. A, B and E are Rrs reflectances. Average is red line and error bars are the standard deviation.**

**Figure 5: Trichodesmium mats detection results on MODIS tile "A2007290.0355": (A) pixels detected with McKinna modified algorithm in red, (B) pixels detected with FAI algorithm, (C) pixels detected with the new algorithm, showing values of Rrs(678) (absolute value).**



**Figure 6: MODIS tile "A2007290.0355" zoomed of the red square area (24°S, Figure 5). Pixels resulting from a false positive detection of (A) the McKinna modified algorithm, (B) FAI and (C) the new algorithm for the area in red on the Figure 5.**

**Figure 7: Retrieval rates vs. distance as a function of the day difference between in situ observations and satellite detection with the new algorithm**

5 **Figure 8: (A) Monthly composite of satellite chlorophyll for March 2015 together with in situ fixation rates superimposed on the OUTPACE track, as colored dots (values on the left colorbar). (B) Points detected as *Trichodesmium* (cyan dots) by the present algorithm together with a summary of absence/presence denoted as colored blue (absence) and red (presence) points along the OUTPACE track. A point at the OUTPACE station is colored when the algorithm shows presence within a 50km radius off the station.**

10 **Figure 9: FAI application to the MODIS tile "A2007290.0355": zoomed of the green square area (22.5 °S, Figure 5). (A) Results at 250 m resolution, (B) the same scene at 1 km resolution. Only a few pixels are remaining corresponding to the densest part of the surface mat, showing the loss of detected mats.**

**Figure 10: MODIS-Aqua image at 250 m taken on March, 6th, 2015 to the south of the OUTPACE cruise illustrating the structure of the chlorophyll (colors) together with the filaments of *Trichodesmium* detected by our algorithm (in black).**

## TABLE LEGEND

**Table 1: Satellite image with in-situ observations used to analyze *Trichodesmium* mat and adjacent spectra.**





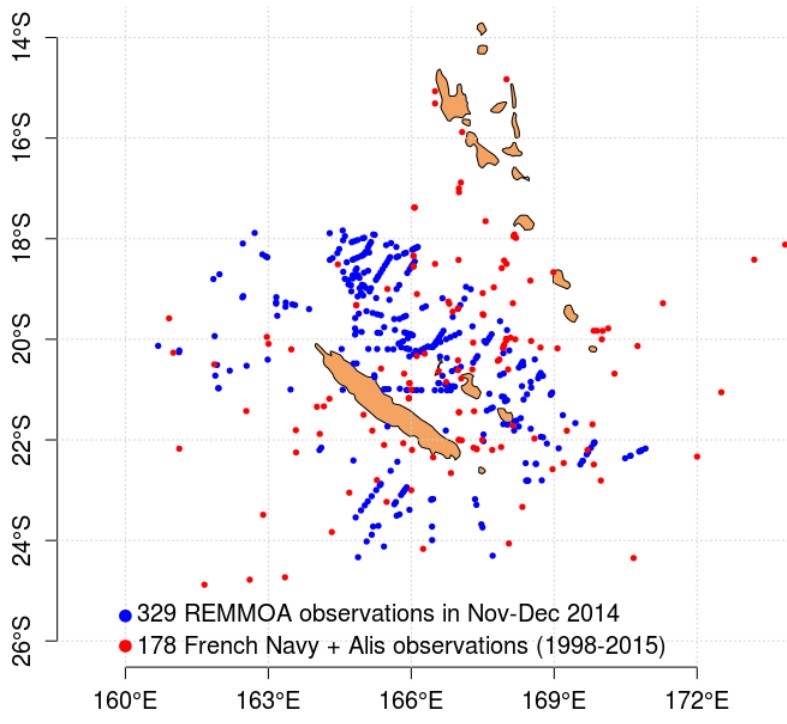

**Fig. 1**





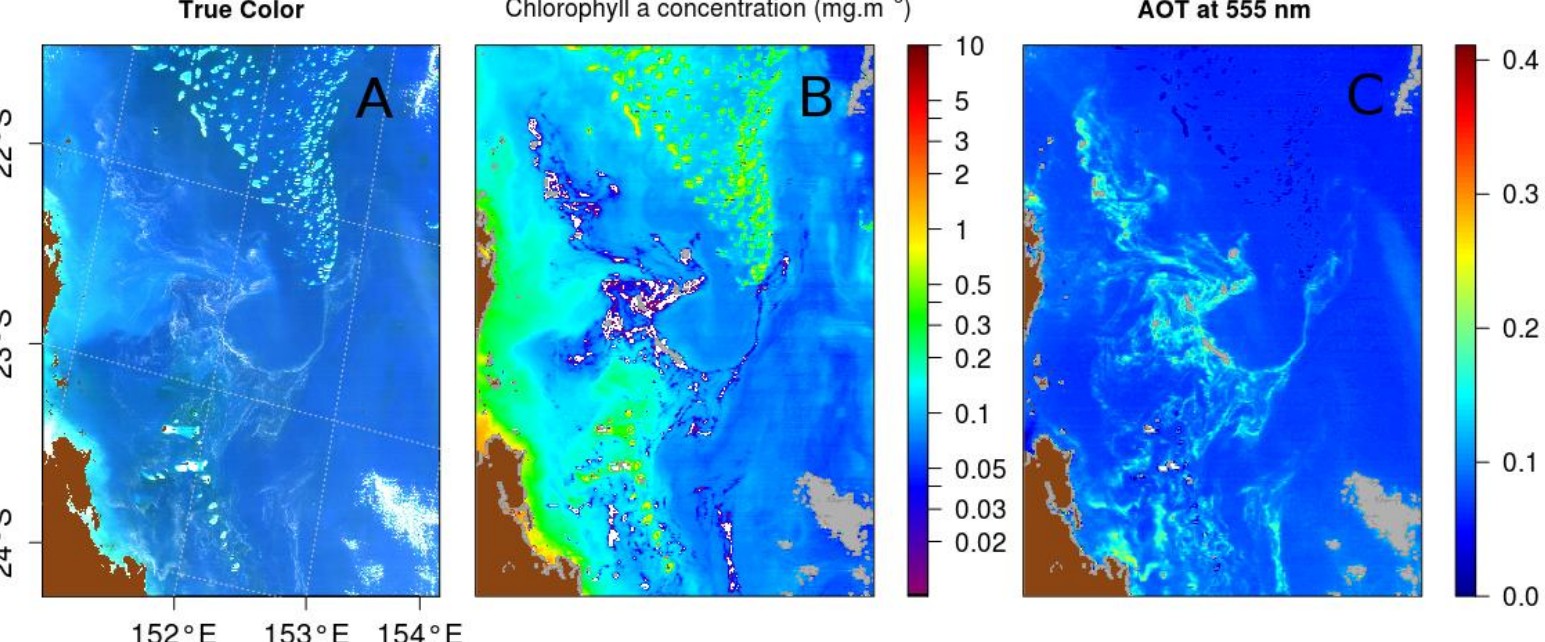

**Fig. 2**





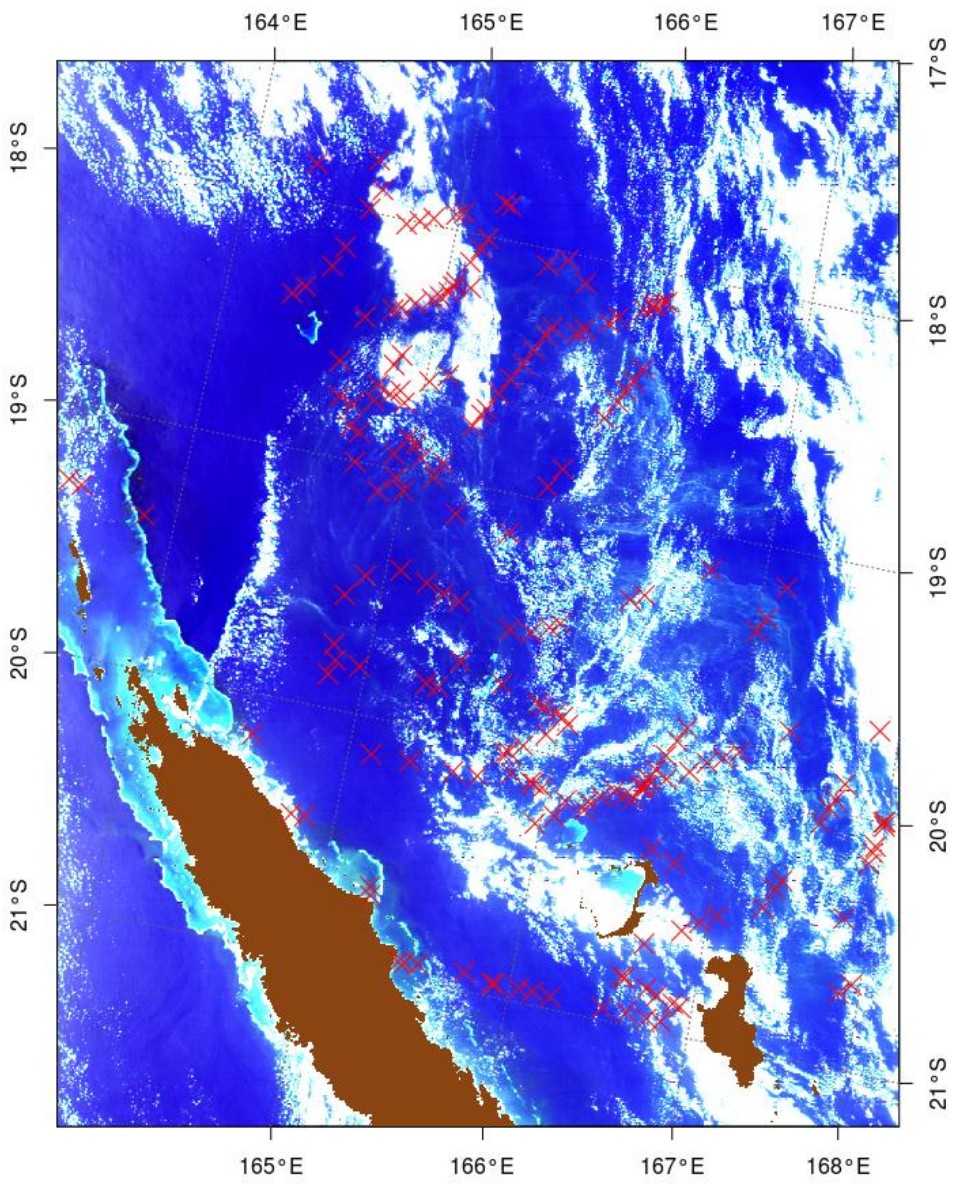

**Fig. 3**





Fig.4





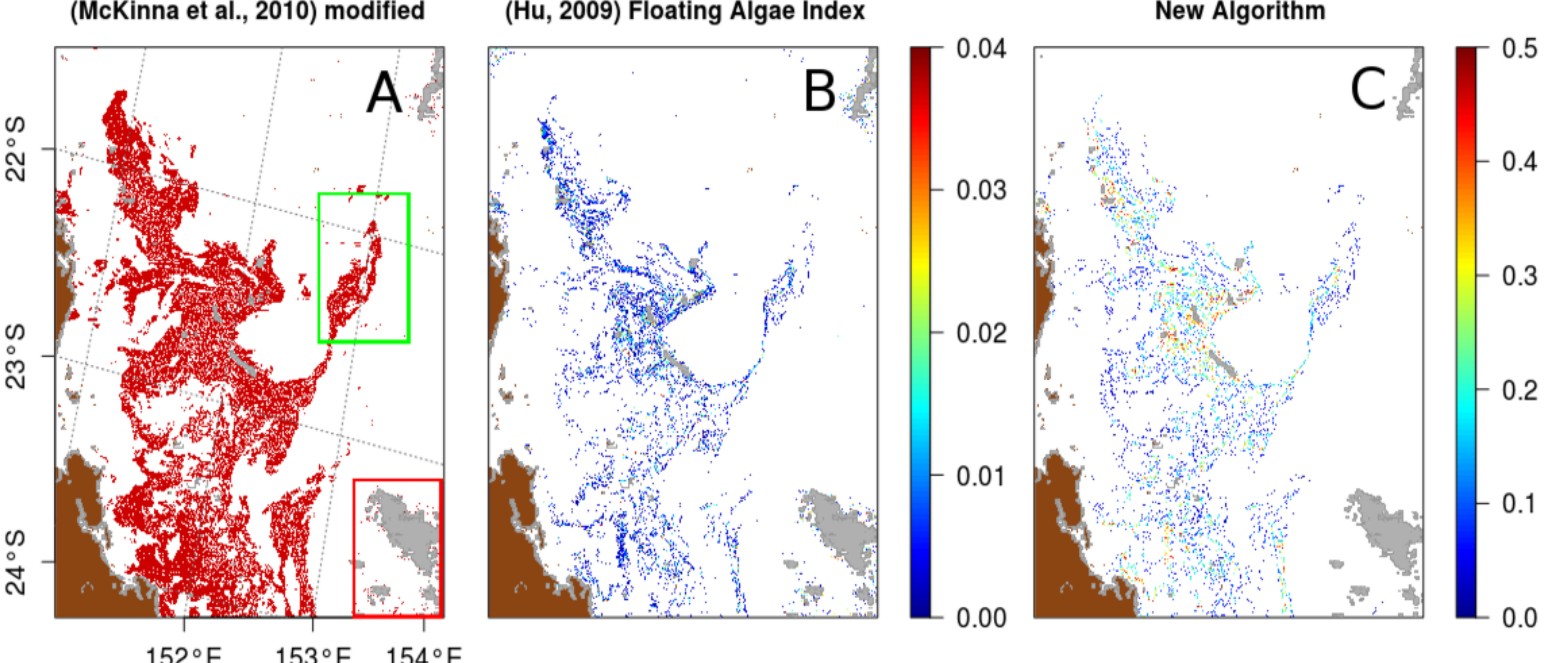

**Fig.5**



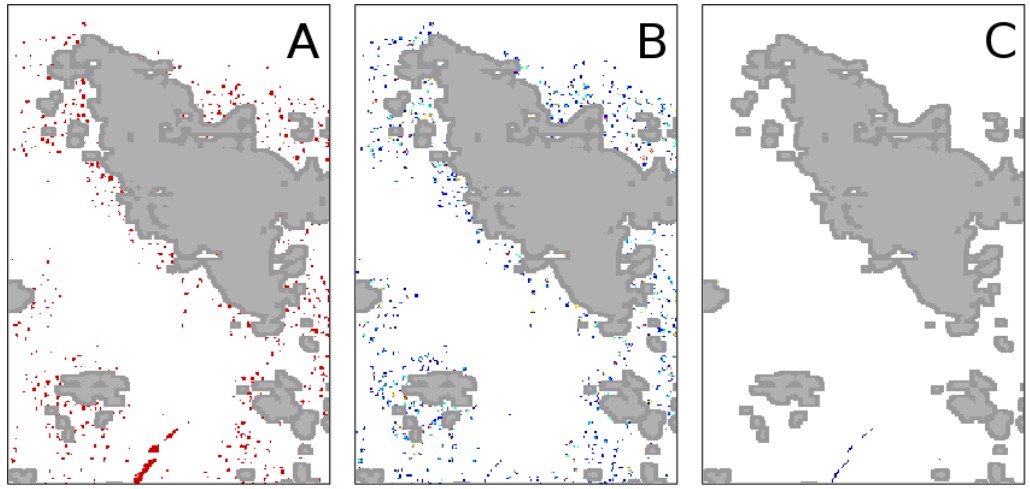

**Fig. 6**





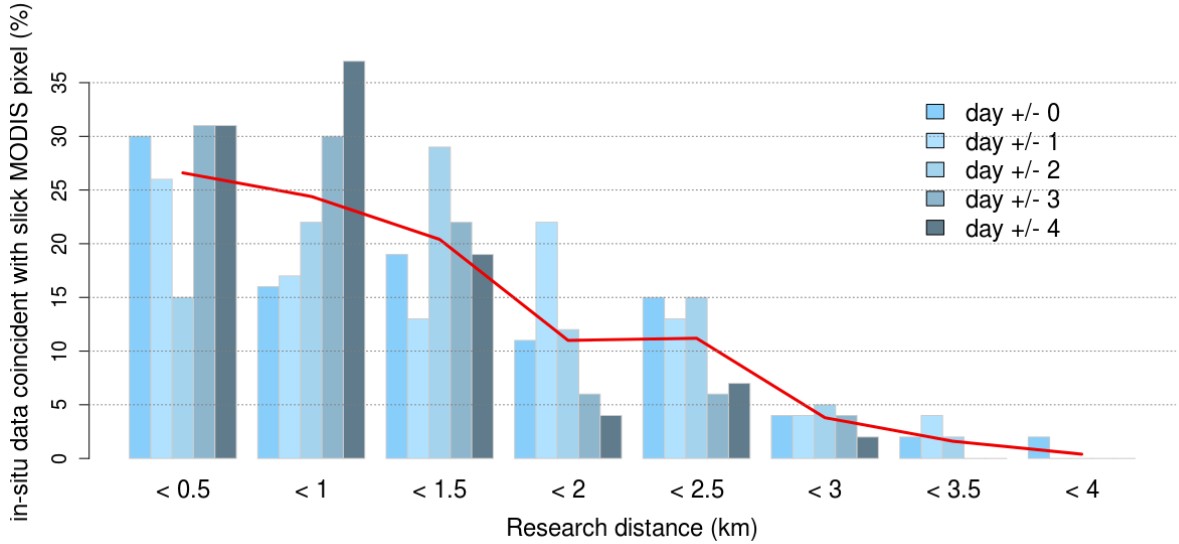

**Fig. 7**







**Fig.8**



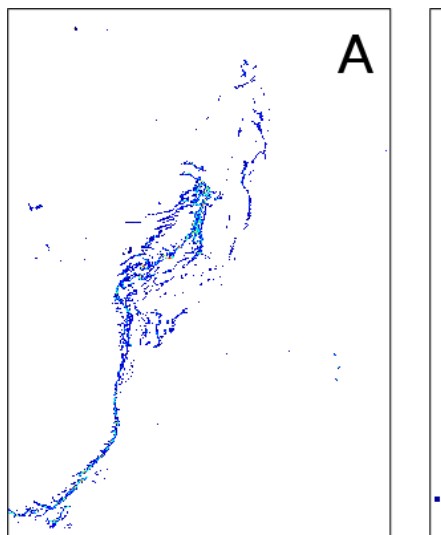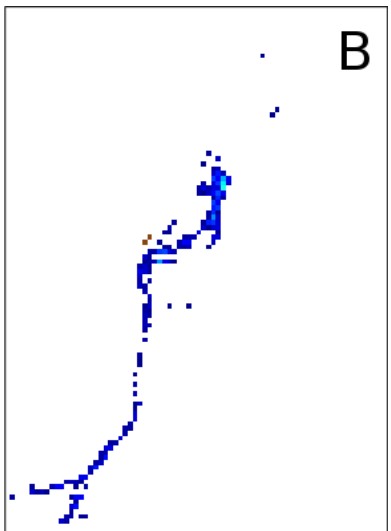

**Fig. 9**



Chlorophyll a concentration (mg.m$^{-3}$)

**Fig. 10**





| Tiles | Date | Location |
|---|---|---|
| A2002341.0255 | 7 December 2002 | East of New Caledonia |
| A2004047.0230 | 16 February 2004 | Loyalty islands |
| A2004059.0255 | 28 February 2004 | East of New Caledonia |
| A2014344.0245 | 10 December 2014 | Loyalty islands (East of Ouvea) |
| A2014351.0255 | 17 December 2014 | Northeast of New-Caledonia |
| A2014353.0240 | 19 December 2014 | Between Vanuatu and New-Caledonia |

5  **Tab. 1**