# Peer review of "Remote Sensing of *Trichodesmium* spp. mats in the Western Tropical South Pacific"

_Biogeosciences, 2017_

## Referee Comment (RC1) · Anonymous Referee #1 · 12 Feb 2018

This is an interesting paper introducing new ship and airborne spotter observations of Trichodesmium and comparing them with MODIS satellite data, but it is at present poorly prepared and written in rather bad English. Text at the start of section 5.1, for example, seems especially confusing and repetitive. Text and logic both need to be made clearer.

The proposed algorithm needs to be better described. Criterion 1 at the bottom of page 7 is dismissed in later text (section 5.1) at "fundamentally a nonsense." This is a bad start. Criterion 2 is said to be concerned with the red edge, but the criterion uses bands at 748 and 859nm, while the red edge is at wavelengths shorter than 748nm, so this cannot be correct. Spectra are shown in Figure 4, and look very similar for Tricho and for "nearby water." A and B show spectra of Tricho mats, but the red edge is

hardly detected. The band at 748nm is not visible in A since its error bars are missing. Similarly for the band at 870nm. All 5 spectra look very similar and detection of Tricho is not made clear.

The authors need to better describe the relevant properties of the MODIS satellite imager. At present, the spatial resolution of 250m is mentioned frequently, even though most spectral bands used have a resolution of 1000m. In both cases, resolution degrades significantly off nadir. Text, for example page 4, lines 15 to 20, seem to suggest that resolution can be chosen and varied for all bands. Text at line 10 on page 9 suggests 250m is the only relevant number. Text at lines 10 to 15 on page 11 expresses it better.

Possible confusion with Sargassum is mentioned at several points in the text. This needs further discussion. Are Sargassum mats commonly/occasionally observed in this area? Have pelagic Sargassum species (Natans or Fluitans) ever been observed in the area?

Something is wrong at the bottom of page 6. MODIS includes all SeaWiFS bands. Gower et al., used MERIS. Was this also a red-edge algorithm?

Smaller points

The study area needs better definition. From its name, the Western Tropical South Pacific must be 0 to 23S, 120E to 180E, which is much larger than the area shown in most Figures.

Excessive use of acronyms makes the paper harder to read. What are LDB, line 25 on page 9, FSLE at the bottom of page 11? Even WTSP is confusing to those of us who do not live there.

Descriptions of Figures needs to be improved. Panels B and C are interchanged in Figure 2. Figure 4 is mentioned above. "Research distance" is a strange variable name in Figure 7. It needs to be explained in the caption. Figure 8 seems to show

Tricho as light blue areas, but the caption refers to cyan dots.

---

## Referee Comment (RC2) · Anonymous Referee #2 · 21 Feb 2018

The study aims at improving the detection of Trichodesmium mats from space with the Ocean Color Sensor MODIS in the South Pacific, in relation with the 2015 OUTPACE campaign. The improvements are seeked in the refinement of the two previously published algorithms of McKinna et al. (2011) and Hu et al. (2010). The authors first analyse the relevance of using Rayleigh corrected radiances Rrc instead of aerosol corrected radiances Rrs. They then adjust empirically the spectral criteria and thresholds of the two algorithms. A collection of closely-matching pairs of satellite detection and in situ sightings or measurements are uses to quantify the performance. The set of adjusted criteria constitutes the new algorithm (as for the previous ones). The matching of the MODIS detection with the in situ detection is statistically satisfying. The new algorithm works better that the previous ones in the vicinity of clouds and leads to fewer

false detections. The authors regret the absence of spectra adquired in situ that could have been compared to the remote sensed spectra in order to explicit the functioning of the new algorithm. The authors acknowledge the possible variation of mat immersion depths as an obstacle to successful detection.

The study is interesting and it is undoubtfully worth improving the detection of this peculiar plankton specie, the abundance of which might increase in a changing ocean. The empirical approach followed is well suited to a very complex remote sensing situation where radiative transfer constrained by (few) spectral measurements can hardly be set up. A considerable amount of work has been done to make the best of an incomplete set of data. A quantification of the performance of the algorithm has been achieved. The relevant litterature is quoted seriously enough.

The study is very worth publishing. The paper itself could however be (formally) improved: - the descriptions of the algorithms and in particular of the new one (will it be given a proper name?) in §3.4 are a bit fastidious and unclear too much. A multi spectral axis diagram could help? - how the tuning was performed? a simple trial and error? - the question of the spectral resolution, which differs from one MODIS band to the other must be explained with more precision and rigor. This should also increase the interest of the comments about the mesoscale spatial structuring of the mats. - the authors might force themselves to split in two as many as possible of their sentences. Beside the gain in fluidity, most of the rather numerous illogicisms should disappear in the process.

Questions:

- The authors write several times that wind mixing could be responsible of the non detection of Tricho. mats. Did they actually met a situation where detection disappeared after a wind event?

- How the immersion effect could be unmixed from the effect of a varying abundance?

A couple of remarks to illustrate style default or lack of precision:

line 31-32 page 6: the sentence does not make sense/ 709 nm instead of 700 nm ? and MERIS instead of SeaWiFS ?

line 15-23 page 12: "MODIS-Terra and MODIS-Aqua satellite sensors are acquiring data since 2000 and 2002 respectively. However, the data quality of these sensors is becoming more and more uncertain with time going by, as their mission was not expected to last more than 6 years. " » mainly the older TERRA is affected. did the authors meet difficulties with AQUA?

" The new algorithm could be adapted to other satellite instruments with similar spectral bands, for example VIIRS onboard NPP and NOAA-20 (1 km resolution) and OLCI onboard Sentinel-3 " » OLCI inherited from MERIS the 709 nm band on which the Gower MCI is based. Are the authors sure that their new algorithm would improve on OLCI-MCI ?

" (300 m spatial sampling), but the spatial 20 resolution remains a problem as we observed that 250 m was already to coarse a resolution to understand the thinner mat dynamics. A study with a better spectral and spatial resolution may lead to better performances and to a new and better algorithm, and this may be possible, at least regarding spatial resolution, with MSI onboard the Sentinel-2 series (10 to 60 m resolution). " » is not the spatial resolution first a problem for the interpretation of the detection itself? (fractional coverage of a pixel by an heterogeneous distribution of algae)

---

## Author Comment (AC1) · 6 Apr 2018

We thank Reviewer 1 for the useful comments provided and address them below.

1) Rev. 1: This is an interesting paper introducing new ship and airborne spotter observations of Trichodesmium and comparing them with MODIS satellite data, but it is at present poorly prepared and written in rather bad English. Text at the start of section 5.1, for example, seems especially confusing and repetitive. Text and logic both need to be made clearer.

Resp.: The text has been changed accordingly, in several places. For example section 5.1., first paragraph:

"Even with a very strong algal concentration, it is possible that with oceanic weather

conditions such as sufficient wind, Trichodesmium scatters and mixes vertically, i.e., we lose the strong signal in the infrared due to the red-edge linked to mats. We are then in the presence of Trichodesmium concentrations that cannot be detected completely with our algorithm. It is successful to locate highly concentrated surface mats, but is not suited for revealing Trichodesmium when scattered under the surface. These are successful to locate the surface mats, but do not succeed in revealing Trichodesmium filaments and/or colonies when they are not aggregated in sea surface mats. We would need, in such situations, a new algorithm, which would allow estimation of Trichodesmium abundance over the whole upper layer. By examining the Rrs spectra of scattered Trichodesmium, obtained during OUTPACE and other cruises, it was not possible to identify clearly characteristics allowing Trichodesmium detection. We find ourselves dealing with a complex problem and a number of variables that, with our current knowledge, do not allow us to create a new bio-optical algorithm and identify robustly Trichodesmium below the surface. [. . .] One should notice that only the densest mats of Trichodesmium are detected with this algorithm. The goal was to provide an algorithm that could detect automatically Trichodesmium in a global scale, and thus limiting the false positive detection as best as possible. Finally, the new algorithm is unable to determine the existence of thin superficial slicks and diffuse Trichodesmium in the water column. Trichodesmium quantification carried out during the OUTPACE campaign (Stenegren et al., 2017) revealed high Trichodesmium abundances near the Fiji island, while our algorithm did not detect them (Figure 8)."

Have been changed into a shorter and clearer version:

"The proposed algorithm was designed to detect strong concentrations of floating Trichodesmium mats and limit wrong detections. However, floating Trichodesmium mats are occurring when sea surface is flat as they tend to sink and disperse for rough conditions (Cecile Dupouy, pers. comm.). In such a case, because of the low penetration depth of NIR irradiance (below 1 m), our algorithm failed to detect sinking Trichodesmium mats even in strong concentration. This situation occurred during

OUTPACE cruise, where measurements reveal high Trichodesmium abundances near the Fiji island (Stenegren et al., 2017), while our algorithm was unable to detect Trichodesmium mats (Figure 8)."

2) Rev. 1: The proposed algorithm needs to be better described. Criterion 1 at the bottom of page 7 is dismissed in later text (section 5.1) at "fundamentally a nonsense." This is a bad start.

Resp.: Because of inappropriate atmospheric correction, near-zero or negative atmospheric corrected reflectances at 678 nm are observed over bloom mats, as already discussed in Hu (2010) and Shanmugam (2011). The result of the overcorrection is indeed "fundamentally a nonsense". By using the 5-min MODIS scene 'granule_id_Mkin', we have tried MUMM (Ruddick et al., 2006), NIR-SWIR (Wang and Shi, 2007) and SWIR (Bailey et al., 2010) more adapted for case 2 waters but we find no improvements and still got negative Rrs. Finally, we used negative Rrs at 678 nm as a convenient threshold to detect bloom mats. To render the algorithm more physical, we also tried to use Rrc at 678 nm only. A new paragraph is added in the discussion section, showing the pros and cons of this simplification.

Hu, C., Cannizzaro, J., Carder, K. L., Muller-Karger, F. E. and Hardy, R.: Remote detection of Trichodesmium blooms in optically complex coastal waters: Examples with MODIS full-spectral data, Remote Sens. Environ., 114(9), 2048–2058, 2010.

Shanmugam, P.: A new bio-optical algorithm for the remote sensing of algal blooms in complex ocean waters, J. Geophys. Res. Ocean., 116(4), 1–12, doi:10.1029/2010JC006796, 2011.

Ruddick Kevin G. , De Cauwer Vera , Park Young-Je , Moore Gerald , (2006), Seaborne measurements of near infrared water‐leaving reflectance: The similarity spectrum for turbid waters, Limnology and Oceanography, 51, doi: 10.4319/lo.2006.51.2.1167.

Menghua Wang and Wei Shi, "The NIR-SWIR combined atmospheric correction approach for MODIS ocean color data processing," Opt. Express 15, 15722-15733 (2007)

Sean W. Bailey, Bryan A. Franz, and P. Jeremy Werdell, "Estimation of near-infrared water-leaving reflectance for satellite ocean color data processing," Opt. Express 18, 7521-7527 (2010)

3) Rev. 1: Criterion 2 is said to be concerned with the red edge, but the criterion uses bands at 748 and 859 nm, while the red edge is at wavelengths shorter than 748 nm, so this cannot be correct.

Resp.: We agree that 748 nm is the upper bound of the red-edge, and the lower bound is <= 700 nm. According to figure 5B of McKinna et al. (2011) and Fig.5 in Hu et al (2010), a positive slope between these wavelengths is observed only over the strongest concentrations of Trichodesmium. Such bloom features (spectral characteristics) have already been pointed by Hu et al. (2010) (high reflectance in NIR (748, 859, and 869 nm)). Rather than using "red-edge" term, we now use "a vegetation effect" in NIR channels only.

Hu, C., Cannizzaro, J., Carder, K. L., Muller-Karger, F. E. and Hardy, R.: Remote detection of Trichodesmium blooms in optically complex coastal waters: Examples with MODIS full-spectral data, Remote Sens. Environ., 114(9), 2048–2058, 2010.

McKinna, L., Furnas, M. and Ridd, P.: A simple, binary classification algorithm for the detection of Trichodesmium spp. within the Great Barrier Reef using MODIS imagery, Limnol. Oceanogr. Methods, 9, 50–66, doi:10.4319/lom.2011.9.50, 2011.

4) Rev. 1: Spectra are shown in Figure 4, and look very similar for Tricho and for "nearby water." A and B show spectra of Tricho mats, but the red edge is hardly detected. The band at 748nm is not visible in A since its error bars are missing. Similarly for the band at 870nm. All 5 spectra look very similar and detection of Tricho is not made clear.

Resp.: By means of the atmospheric corrections at 859 nm and 748 nm, Rrs at these

wavebands are set to zero (explaining the lack of error bars). However it is still possible to observe strong values at 859 nm and 748 nm in the RRc spectrum (Figure 4B). In addition to a negative value for the Rrs (or the clear trough for the Rrc at 678 nm) over mats, the difference between Trichodesmium mat spectrum and "nearby water" (Figure 4C-D) is, according to us, quite visible. The error bar of the figure have been expanded to be more visible. The water spectrum has also been added, helping the comparison between the spectra. Additional information are indicated on the figure for a better comprehension. The legend and the text explaining the figure have been changed accordingly.

5) Rev. 1: The authors need to better describe the relevant properties of the MODIS satellite imagery. At present, the spatial resolution of 250m is mentioned frequently, even though most spectral bands used have a resolution of 1000m. In both cases, resolution degrades significantly off nadir. Text, for example page 4, lines 15 to 20, seem to suggest that resolution can be chosen and varied for all bands. Text at line 10 on page 9 suggests 250m is the only relevant number. Text at lines 10 to 15 on page 11 expresses it better.

Resp.: The text has been changed to reflect this (section 2.2), see below. Moreover an additional table (Table 1) has been added to show the different bands used, their resolution and their key use by NASA Ocean Biology Processing Group (OBPG).

"We used MODIS atmospheric corrected (aerosol+Rayleigh) reflectances (Rrs) in visible, near-infrared (NIR) and short wavelength infrared (SWIR) at different resolutions: 250 m resolution for bands 1 (645 nm) and 2 (859 nm), 500 m resolution (bands 3-7, visible and SWIR land/clouds dedicated bands), and 1 km resolution (bands 8-16). Bands 8 to 16 are dedicated ocean color bands (Table 1), but we also use information in high-resolution bands located in the visible-NIR region to track floating blooms. To evaluate the influence of resolution on detection performances, Level-2 remote sensing data was produced at both 250 m and 1 km resolutions, with interpolation of 500 m and 1 km channels and aggregation of 250 m resolution channel respectively. The

consequences of these processing are discussed in Section 5. "

6) Rev. 1: Possible confusion with Sargassum is mentioned at several points in the text. This needs further discussion. Are Sargassum mats commonly/occasionally observed in this area? Have pelagic Sargassum species (Natans or Fluitans) ever been observed in the area?

Resp.: To our knowledge Sargassum (natans, fluens) form rafts in open ocean waters only in the Atlantic Ocean. Pelagic Sargassum species were never observed in the studied area (Payri and Richer de Forges, 2000). Sargassum rafts have only been observed in lagoons of French Polynesia (S. polycystum; Andrefouet et al., 2017).

Payri C. and B. Richer de Forges, 2000. Compendium of marine species from New Caledonia. ISSN 1297-9635, Second edition, N° 117, ORSTOM editions, IRD Center of Noumea, 480 pages.

Andréfouët S., Payri C., Van Wynsberge S., Lauret O., Alefaio S., Preston G., Yamano H., Baudel S. The timing and the scale of the proliferation of Sargassum polycystum in Funafuti Atoll, Tuvalu. Journal of Applied Phycology, 29 (6), 3097-3108 (2017).

7) Rev. 1: Something is wrong at the bottom of page 6. MODIS includes all SeaWiFS bands. Gower et al., used MERIS. Was this also a red-edge algorithm?

Resp.: Gower et al. in his algorithm use a little fluctuation near 709 nm to detect Trichodesmium. However in MODIS this fluctuation is not present because the band-width is much larger (650-700 nm) and thus cannot be used. The mistake about using SeaWIFS instead of MERIS for the Gower reference has been corrected.

8) Rev. 1: The study area needs better definition. From its name, the Western Tropical South Pacific must be 0 to 23S, 120E to 180E, which is much larger than the area shown in most Figures. Excessive use of acronyms makes the paper harder to read. What are LDB, line 25 on page 9, FSLE at the bottom of page 11? Even WTSP is confusing to those of us who do not live there.

Resp.: Correction made. The acronyms has been explicitly described. The WTSP refers to the Western Tropical South Pacific and is the OUTPACE area which had been set as the default area for the various papers of this special edition.

9) Rev. 1: Descriptions of Figures needs to be improved. Panels B and C are interchanged in Figure 2. Figure 4 is mentioned above. "Research distance" is a strange variable name in Figure 7. It needs to be explained in the caption. Figure 8 seems to show Tricho as light blue areas, but the caption refers to cyan dots.

Resp.: The changes in Figure 4 are mentioned above. Others figures have been updated in accordance to this comment.

---

## Author Comment (AC2) · 6 Apr 2018

We thank Reviewer 2 for the useful comments provided and address them below.

1) Rev. 2: The study is very worth publishing. The paper itself could however be (formally) improved: - the descriptions of the algorithms and in particular of the new one (will it be given a proper name?) in §3.4 are a bit fastidious and unclear too much. A multispectral axis diagram could help?

To help the reader, a new section '3.3. Robust spectral features over and near Trichodesmium mats' has been included before section 3.4. Section 3.3. aims at giving details on the main spectral characteristics over floating blooms. We also managed to make the Figure 4 clearer, by comparing with spectra over blue water, used as reference.

2) Rev. 2: How the tuning was performed? a simple trial and error?

Resp.: First, a bunch of MODIS spectra over Trichodesmium mats, adjacent and over blue waters pixels as a reference was inspected to detect spectral characteristics of each group. In section 3.3, we discuss the fact that Tricho detection can be achieved by 2 main criteria: a trough in the spectrum at 678nm as well as a vegetation NIR effect at 859nm resulting in positive spectral slope between 748nm and 859nm. We used these two conditions. A condition equivalent to a trough at 678nm in Rrc is Rrs (678 nm) <0 (see section 3.3 for more explanation). Hence the use of the latter condition (equation 1) as well as equation 2 for the NIR. To complete the algorithm, we searched a criteria avoiding false positive due to remaining cloud edges after cloud masking. The best criteria found to eliminate those is described in equation 3. That criteria may be understood by the fact that at these cloud edge pixels, a mixture of water and cloud signals result empirically in a positive spectral slope between 531 and 645nm. This justifies empirically the use of criterium 3.

3) Rev. 2: - the question of the spectral resolution, which differs from one MODIS band to the other must be explained with more precision and rigor. This should also increase the interest of the comments about the mesoscale spatial structuring of the mats.

In accordance to the reviewer's comment, a new table detailing MODIS bands used in this study has been provided. Section 2.2. on Satellite data has been improved as indicated in our response to Reviewer 1's comment 5), see above.

If the reviewer means spatial resolution (because of his/her mention of spatial restructuring), over Trichodesmium mats of few tens of meter large, the 748nm band (1 km resolution) would take much more surrounding water into account than the 859 nm band (250 m resolution). Therefore, even with spatial interpolation of the 748nm band to 250m, the reflectance value in this band remains lower than that at 859 nm. Hu et al. (2010) came to similar conclusions. This explanation was added to the manuscript.

Now regarding spectral resolution, a better spectral resolution (i.e., smaller bandwidth) in the 645 and 859 nm bands, which would favor longer wavelengths in the first band and shorter wavelengths in the second band, would increase the difference between Rrc at 645 and 531 nm and 748 and 859 nm for a better detection. The following sentence was added at the end of Section 3: "Note that a better spectral resolution (i.e., smaller bandwidth) for the 645 and 859 bands, keeping the longest and shorter wavelengths, respectively, would enhance the differences between Rrc at 645 and at 531 nm and Rrc at 748 and at 859 nm, facilitating Trichodesmium detection."

4) Rev. 2: - the authors might force themselves to split in two as many as possible of their sentences. Beside the gain in fluidity, most of the rather numerous illogicisms should disappear in the process.

Resp.: This was also requested by the first reviewer. The text has been changed to be clearer, see Reviewer 1, comment 1).

5) Rev. 2: - The authors write several times that wind mixing could be responsible of the non detection of Tricho. mats. Did they actually met a situation where detection disappeared after a wind event?

The situation where detection disappeared after a wind event was frequently observed in the open ocean around New Caledonia and also inside the lagoon. During the 9 Diapalis (DIAzotrophy in the PAcific zone with ALIS) cruises around New Caledonia, slicks were observed only in October 2001. Though in February 2003, abundance was 5000 trichomes /L, no slick was observed due to strong wind (25 knots) (Tenorio et al., 2018). At the inverse, in October 2003, thin slicks of Trichodesmium were observed during extremely calm conditions, while global abundance was not very high.

Tenorio, M., Dupouy, C., Rodier, M., Neveux, J.: Trichodesmium and other Filamentous Cyanobacteria in New Caledonian waters (South West Tropical Pacific) during an El Niño Episode, Aquatic Microbial Ecology, 2018
6) Rev. 2: - How the immersion effect could be unmixed from the effect of a varying abundance?

The immersion effect is a major problem for slick and mat detection. Indeed, the interpretation of Remote sensing reflectances (Rrs) is complicated by the vertical structure of inherent optical properties in the ocean surface layer. The influence on reflectance of a thin deep layer (at 2.5 m) versus a surface layer has been discussed in Petrenko et al. (1998). They conclude that the deeper the layer, the more unreliable the estimation of the chlorophyll concentration derived with algorithms based on reflectance ratios. For example, during Diapalis cruises, deep chlorophyll peaks measured in-situ due to thin layers of Trichodesmium colonies observed at 5 and 10 meters were associated with no or low surface abundance derived using ocean color algorithms. Having said that, we agree that at this stage it is not possible to distinguish between the two effects. In this article, we concentrate on the special case of surface accumulations by convergence of winds or currents. Our algorithm is not suited for detection of subsurface accumulations.

Petrenko, A.A., J.R.V. Zaneveld, W.S. Pegau, A.H. Barnard, and C.D. Mobley. 1998. Effects of a thin layer of reflectance and remote-sensing reflectance. Oceanography 11(1):48–50, https://doi.org/10.5670/oceanog.1998.15.

7) Rev. 2: line 31-32 page 6: the sentence does not make sense/ 709 nm instead of 700 nm ? and MERIS instead of SeaWiFS ?

Resp.: The confusion between SeaWiFS and MERIS has been corrected, as well as the wavelength used in the MCI algorithm by Gower et al. (2014).

8) Rev. 2: line 15-23 page 12: Âń MODIS-Terra and MODIS-Aqua satellite sensors are acquiring data since 2000 and 2002 respectively. However, the data quality of these sensors is becoming more and more uncertain with time going by, as their mission was not expected to last more than 6 years. " Âż mainly the older TERRA is affected. did the authors meet difficulties with AQUA?

It is well known that MODIS/Terra (and Aqua) is suffering from multiple issues detailed by Franz et al. (2008), but efforts were conducted by the NASA OBPG to provide Ocean Color retrievals based on Terra, to complete the time serie from Aqua. Cross-calibration operations and validation of MODIS /Aqua and /Terra reflectances with in-situ data are periodically updated by NASA OBPG. For example, results from the last R2018.0 re-processing of MODIS/Terra show that Terra reflectances compare well with in-situ observation (see https://oceancolor.gsfc.nasa.gov/reprocessing/r2018/terra/). We were not concerned by issues affecting the quality of water-leaving radiances derived from Aqua and Terra, and we did not make any comparison between Aqua and Terra to judge the detection performance of Trichodesmium.

Byran A. Franz, Ewa J. Kwiatowska, Gerhard Meister, Charles R. McClain, "Moderate Resolution Imaging Spectroradiometer on Terra: limitations for ocean color applications," Journal of Applied Remote Sensing 2(1), 023525 (1 June 2008). https://doi.org/10.1117/1.2957964

9) Rev. 2: " The new algorithm could be adapted to other satellite instruments with similar spectral bands, for example VIIRS onboard NPP and NOAA-20 (1 km resolution) and OLCI onboard Sentinel-3 " Âż OLCI inherited from MERIS the 709 nm band on which the Gower MCI is based. Are the authors sure that their new algorithm would improve on OLCI-MCI ?

Resp.: This algorithm has been created with the automatization process for MODIS in mind. The algorithm is tuned to avoid false positive and likely underestimates the Trichodesmium abundance compared to others, such the Gower MCI based on radiances at 709 nm with spatial resolution 300m or OLCI with similar resolution.

10) Rev. 2: " The 300 m spatial sampling of Sentinel 3 would be perfectly adapted to the detection of mats as we do it on the 250m-resolution MODIS channels. Indeed, studies of small parts of the mats with a better spectral and spatial resolution may lead to better performances with MSI onboard the Sentinel-2 series (10 to 60 m resolution).

"ż is not the spatial resolution first a problem for the interpretation of the detection itself? (fractional coverage of a pixel by an heterogeneous distribution of algae).

Resp.: As we understand the question, indeed the spatial resolution is the first problem when detecting Trichodesmium. Figure 4 shows the difference in the detection between 1 km and 250 m resolution. At 1 km the Trichodesmium signal is barely recognizable from the water signal. Moreover as discussed in this article the Trichodesmium mat width does not exceed 50 m, and with the decrease of the spatial resolution, the Trichodesmium signal is drown in the water signal. Nevertheless more there are spectral band to analyse the signal better it is. However, if a better spatial resolution is foremost important, a better spectral resolution (e.g., hyper-spectral measurements) would allow, for example, a better detection of the red edge. We deleted "spectral and" and ", at least regarding spatial resolution." in the second sentence of the text quoted by the reviewer.

---

## Author Response (AR2)

*1) Rev.: My major criticism is the apparent lack of a clear detection of the expected spectral signature of Trichodesmium slicks, which show a strong red edge, as described in section 3.3. A strong red edge is shown in Gower et al., 2014, Figure 6 derived from MERIS, and this should also appear in MODIS data. Spectra in Figure 4 show little sign of a red edge.*

Resp: The values shown Figure 4 is the average of 600 differents values taken from different Trichodesmium mat density. The majority of them present a weak response in the red-edge, but are nevertheless Trichodesmium algae agglomeration. The densest mats have a stronger reflectance in the NIR. We can see these difference with the in situ spectrum taken by McKinna et al. (2015), Figure 3.

McKinna, L., Furnas, M., and Ridd, P.: A simple, binary classification algorithm for the detection of Trichodesmium spp. within the Great Barrier Reef using MODIS imagery, Limnology and Oceanography: Methods, 9, 50–66, https://doi.org/10.4319/lom.2011.9.50, 2011.

*2) Rev.: The paper needs work. English is poor. There are several typos.*

Resp: A read-through of the article have been carried out to limit typos.

*3) Rev.: Line 27 of page 7 mentions the need to distinguish Sargassum in this area. How realistic is this? Has Sargassum ever been observed there in quantities large enough to cause confusion?*

Resp: To today, no important Sargassum bloom have been observed in the Southwestern Pacific (Payri and Richer de Forges, 2000). If the second step of the algorithm of Hu et al. (2010) resolve the ambiguity between Trichodesmium and Sargassum, this distinction have not been studied with the new algorithm.

Payri C. and B. Richer de Forges, 2000. Compendium of marine species from New Caledonia. ISSN 1297-9635, Second edition, N°117, ORSTOM editions, IRD Center of Noumea, 480 pages.

Hu, C., Cannizzaro, J., Carder, K. L., Muller-Karger, F. E. and Hardy, R.: Remote detection of Trichodesmium blooms in optically complex coastal waters: Examples with MODIS full-spectral data, Remote Sens. Environ., 114(9), 2048–2058, 2010.

*4) Rev.: A possible daily cycle in Trichodesmium observability is mentioned on page 9. This could be important and needs more discussion. How big a variation has been observed? How consistently?*

According to Villareal and Carpenter, *Trichodesmium* colonies migrate vertically in the water column. To our knowledge, no study of the abundance variation of *Trichodesmium* between these cycles have been carried out. Page 9, the main point was around the vertical migration of *Trichodesmium* because of natural or environmental causes that may affect the detection of the algorithm.

Villareal, T. A. and Carpenter, E. J.: Buoyancy Regulation and the Potential for Vertical Migration in the Oceanic Cyanobacterium Trichodesmium, Microbial Ecology, 45, 1–10, http://www.jstor.org/stable/4287673, 2003.

*5) Rev.: The MCI time sequence in Gower et al., 2014 shows monthly composite images over this area for the period 2002 to 2012 (Figure2 in that paper). Do any of the high signals shown there correspond to in-situ data considered here?*

Resp: For the area considered in Gower et al. and the period 2002-2012, only 80 observations remain. 45% of these observations are coincident with the monthly composite images of Gower et al., 2014. The *Trichodesmium* blooms during the period Jan-Feb 2003, Feb 2004, Mar-Apr and Dec 2010 are especially interesting and match with 27 of our observation. For the observations not detected, 30 were observed in April and July 2003.

*6) Rev.: Line 5. The new algorithm discussed in section 3.5 uses negative Rrs(678) to measure a red-edge signal. Using the absolute value of Rs(678) will result in errors if Rs(678) is ever positive. The negative value alone should be used.*

Resp: The reviewer is correct, actually the detection does not use the magnitude of Rrs(678). The absolute value is only used when negative. The sentence has been reformulated to remove the confusion.

**Minor points:**
**Page 1.**
*7) Rev.: Line 1 "is the major dinitrogen-fixing organisms." Should be "organism." Do we need the "di"? Are there "mono-nitrogen" fixers?*

Resp: "dinitrogen-fixing" have been changed into "nitrogen-fixing"

*8) Rev.: Line 15, "fails to detect sub-surface booms." This should be "blooms." But why criticise the algorithm for not being able to do something which must be fundamentally impossible? There can be no signal from (deeper) sub-surface blooms.*

Resp: This sentence states the limit of the be detection for reader which would not be familiar with this kind of remote sensing algorithm. The sentence has been deleted nevertheless.

**Page 3**
*9) Rev.: Lines 11,12, "blooming period (Nov to Mar) coincides with the South Pacific Convergence Zone." A period has to coincide with another period, or Zone with another Zone. I'm not sure what point you are making here.*

Resp: The South Pacific Convergence Zone (SPCZ) is an area of convergence with an important cloud cover. The SPCZ does not have a specific location and move between seasons. Here, during the period from November to Mars, the SPCZ intersect our zone of study, the WTSP. Formulation corrected according to comment.

*10) Rev.: Line 32, "three datasets intersecting." Maybe you mean "three datasets providing data in the acquisition period"*

Resp: The reviewer is correct. Formulation corrected according to comment.

**Page 4**

*11) Rev.: Lines 17,18 "SOB database and the OUTPACE campaign are used. So REMMOA is not used? Why not? You discuss it above and call it "most favorable for satellite data validation" (line 6).*

Resp: As discussed before, in section 2.1, the REMMOA dataset is included in the SOB database . As it did not seem clear that SOB integrated REMMOA, the paragraph was reformulated to make it clearer.

**Page 10**
*12) Rev.: Lines 15, 16. Where in Figure 7 is the area you discuss?*

Resp: The area is just around the stations near the longitude 180° at the South of Fidji of the OUTPACE campaign. This information has been added to the text.

**Page 11**
*13) Rev.: Line 5. "Two adjacent wavelengths 645 and 748." The two adjacent wavelengths are 665 and 748. Why use 645 in Equation 4?*

Resp: Indeed the wording was not correct, as 645 is not adjacent to 678. The choice of 645 instead of the adjacent wavelength allows to estimate the trough at 678 nm in the spectrum (Figure 4A) which would not be as clear when using the adjacent wavelength at 665 nm. The wording have been corrected.

*14) Rev.: What is the criterion in equation 4? Must this expression be positive? Or must it be greater than some threshold value?*
Resp: You're right, there was a missing criteria in the version you had for equation 4. It has been corrected in the new version (Equation 4 must be greater than 0).

*15) Rev.: Line 15,16 Which "former one"?*

Resp: The algorithm proposed section 3.5. The text have been changed accordingly

**Page 12**
*16) Rev.: Line 6. What is sub-mesoscale? By one convention mesoscale is about 300km, but you must mean a much smaller value than this.*

Resp: Lévy et al. 2012 name "submesoscales" scales between 1 and 10 km at most and "mesoscales" is used for scales of the order of the first baroclinic radius of deformation. Typically at 20° S these scales are ~50 km (e.g. http://www-po.coas.oregonstate.edu/research/po/research/rossby_radius/fig2.html). We wanted to emphasize that visual observations at sea typically show sub mesoscales structures that cannot be fully described by > 12.5km FSLEs as used by Rousselet et al (2018).

Lévy, M., R. Ferrari, P. J. S. Franks, A. P. Martin, and P. Rivière (2012), Bringing physics to life at the submesoscale, *Geophys. Res. Lett.*, 39, L14602, doi: 10.1029/2012GL052756.

**Page 13**
*17) Rev.: Line 21. "citetMcKinna2011" should be "cited McKinna (2011)"*

Resp: corrected

**Page 14**

*18) Rev.: Line 7 "floating algae organized in a heap" is a strange description. Perhaps you mean "floating algae in mats of varying surface concentration"?*

Resp: The description has been changed, we meant "organized in dense mats".

**Figure 2**
*19) Rev.: Panels B and C captions are interchanged*

Resp: To our understanding the captions between panel B and C are not interchanged. Panel B is the spectra of Trichodesmium mats at 1 km resolution with $R_{rc}$ reflectances. Panel C is Trichodesmium mats at 250 m resolution with $R_{rs}$ reflectances.

**Figure 4**
*20) Rev.: Why are points not plotted at 748 and 870nm in panels A, C and E?*

Resp: Because of the atmospheric correction used in SeaDas, these two wavelength are set to 0. Details Section 3.3.

**Figure 5**
*21) Rev.: The geographic area needs to be better described. I assume this is the Great Barrier Reef, with land at bottom left and cloud in grey? It would be useful to outline the area of the reef itself.*

Resp: The label of the Great Barrier Reef have been added Figure 2 and Figure 5, and the caption of the figure have been completed to describe the grey pixel as clouds.

**Figure 6**
*22) Rev.: The grey areas appear to be clouds. This needs to be made clear.*

Resp: The caption of the figure have been completed to describe the grey pixel as clouds.

**Figure 7**
*23) Rev.: The Figure needs better description. Presumably, A is the top panel, B the bottom one? What are the units of the two color bars? At present the left color bar is referred to as fixation rate, the central bar is undefined. Text describing the colored dots in the lower panel is ambiguous: they are said to be blue for absence, red for presence, but the last sentence says they are only colored for presence.*

Resp: The figures have been labelled, the central bar have been defined, and the caption have been made clearer.

**Figure 8**
*24) Rev.: "Only a few pixels are remaining." "Only a few pixels remain" is better English. In fact, I would say that a significant number remain, or "The main feature is well-detected at the lower resolution."*

Resp: The sentence have been corrected

[revised manuscript text omitted]